# Metal-Coordinated Nanofiltration Membranes Constructed on Metal Ions Blended Support toward Enhanced Dye/Salt Separation and Antifouling Performances

**DOI:** 10.3390/membranes12030340

**Published:** 2022-03-18

**Authors:** Xiaofeng Fang, Shihao Wei, Shuai Liu, Ruo Li, Ziyi Zhang, Yanbiao Liu, Xingran Zhang, Mengmeng Lou, Gang Chen, Fang Li

**Affiliations:** 1Textile Pollution Controlling Engineering Centre of Ministry of Ecology and Environment, College of Environmental Science and Engineering, Donghua University, Shanghai 201620, China; wei20211226@163.com (S.W.); liushuai5274@163.com (S.L.); 18379467166@163.com (R.L.); zzy13966616646@163.com (Z.Z.); yanbiaoliu@dhu.edu.cn (Y.L.); xrzhang@dhu.edu.cn (X.Z.); mengmeng_lou@outlook.com (M.L.); cheng@dhu.edu.cn (G.C.); 2Key Laboratory of New Membrane Materials, Ministry of Industry and Information Technology, Nanjing University of Science & Technology, Nanjing 210094, China; 3Shanghai Institute of Pollution Control and Ecological Security, Shanghai 200092, China

**Keywords:** nanofiltration, metal-coordination, polyphenol, dye/salt separation, antifouling

## Abstract

Metal-phenol coordination is a widely used method to prepare nanofiltration membrane. However, the facile, controllable and scaled fabrication remains a great challenge. Herein, a novel strategy was developed to fabricate a loose nanofiltration membrane via integrating blending and interfacial coordination strategy. Specifically, iron acetylacetonate was firstly blended in Polyether sulfone (PES) substrate via non-solvent induced phase separation (NIPS), and then the loose selective layer was formed on the membrane surface with tannic acid (TA) crosslinking reaction with Fe^3+^. The surface properties, morphologies, permeability and selectivity of the membranes were carefully investigated. The introduction of TA improved the surface hydrophilicity and negative charge. Moreover, the thickness of top layer increased about from ~30 nm to 119 nm with the increase of TA assembly time. Under the optimum preparation condition, the membrane with assembly 3 h (PES/Fe-TA3h) showed pure water flux of 175.8 L·m^−2^·h^−1^, dye rejections of 97.7%, 97.1% and 95.0% for Congo red (CR), Methyl blue (MB) and Eriochrome Black T (EBT), along with a salt penetration rate of 93.8%, 95.1%, 97.4% and 98.1% for Na_2_SO_4_, MgSO_4_, NaCl and MgCl_2_ at 0.2 MPa, respectively. Both static adhesion tests and dynamic fouling experiments implied that the TA modified membranes showed significantly reduced adsorption and high FRR for the dye solutions separation. The PES/Fe-TA3h membrane exhibited high FRR of 90.3%, 87.5% and 81.6% for CR, EBT and MB in the fouling test, stable CR rejection (>97.2%) and NaCl permeation (>94.6%) in 24 h continuous filtration test. The combination of blending and interfacial coordination assembly method could be expected to be a universal way to fabricate the loose nanofiltration membrane for effective fractionation of dyes and salts in the saline textile wastewater.

## 1. Introduction

Due to the rapid development of the textile industry, a large amount of wastewater is produced and discharged [1,2]. The textile wastewater generally consists of dyes, inorganic salts and other chemicals [3]. The discharge of such textile wastewater has negative effects on aquatic ecosystems and public health due to the features of highly toxic and bio-accumulation of dyes [4,5]. It is noteworthy that the existence of salt impedes textile wastewater from biodegrading. In addition, these inorganic salts are also a recyclable resource in textile wastewater [6]. Therefore, separating salt/dye mixture is a critical step to reuse the inorganic salts and polluted textile wastewater [7].

Membrane separation technology is deemed to be an effective way for treating textile wastewater, owing to its small footprint, low energy consumption and high selectivity [8,9,10,11,12,13]. Typically, nanofiltration (NF) has become one optimal choice for removing organic matters with a molecular weight of 200–1000 Da. However, most commercial NF membranes with a dense separation layer exhibit high salts rejection and low permeability [14,15]. Thus, it is unsatisfactory for separating dye/salt in textile wastewater to recycle the resources. To overcome this problem, the loose nanofiltration membrane (LNM) has recently drawn intense attention to achieve the effective fractionation of dye and salt [16,17,18,19]. Compared with traditional NF membranes, LNMs possess a relatively looser structure and larger pore size, which promote water and salt permeation. The organic dyes are rejected by the combination of size exclusion and electrostatic repulsion [20,21]. For the dyeing wastewater treatment, LNMs have high efficiency and economic value for separating organic pollutants from inorganic salts [3,22,23]. During the past few years, various LNMs and their separation mechanisms have been reported [24,25,26]. The pore size distribution and surface properties of the separation layer are pivotal factors to enhance the separation efficiency for the dye/salt mixture. In view of the academic and practical development of LNMs, the rational design and precise manipulation of the separation layer is quite challenging.

Polyphenol chemistry, including metal-phenol coordination and amine-phenol deposition, has received significant attention as effective tools for the preparation of the separation membrane [27,28,29]. Tannic acid (TA), a natural plant polyphenol with abundant catechol and pyrogallol groups, has been widely used in the preparation and modification of membranes, because of the presence of abundant active groups and low-cost properties [30]. The catechol and pyrogallol groups can crosslink with metal ions and organic molecules to form the complex. The application of a TA-based complex for membrane separation has been investigated in some meaningful research works [31,32]. For instance, Li et al. [33] have demonstrated the co-deposition of TA and PEI to prepare LNF membrane-selective layers. Shao et al. [34] developed the novel metal-TA network to prepare high-flux nanofiltration membranes. Wu et al. [35] have fabricated a low-pressure nanofiltration membrane via a bio-inspired one-pot assembly on the polysulfone (PSf) substrate with a tannic acid-titanium (TA-Ti) network coating as the selective layer. Fan et al. [36] have reported the preparation of an LNF membrane via coordination complexes of TA and iron (III) ions. Peinemann et al. [37] reported a facile and cost-effective co-deposition method to prepare NF membranes via the complexation of TA and copper. Shen et al. [38] used the TA and Fe as the two-phase monomers to fabricate metal-organic composite membrane via the interfacial coordination method. The obtained LNMs in the above-mentioned works exhibited both high dye rejection and salt permeation. However, these preparation methods, such as co-deposition process and biphasic interfacial coordination, are uncontrollable due to the rapid reactions in the mixed system and the aggregated particles are easy to form on the substrate surface. In addition, the stability of the separation layer should be considered, since that there is no direct chemical bonding between the TA layer and the substrate in most systems. Additionally, while most of these studies were performed at the laboratory scale, the scale-up production of NF membranes with polyphenol chemistry is still difficult. Therefore, a facile, precision controllable and widely applicable strategy for the LNM construction with precise dye/salt separation is still needed.

In this work, a novel strategy was developed to fabricate a loose separation layer via integrating blending and interfacial coordination. For the metal-phenol coordination coating, the metal ion source is a critical point. Introduction of the metal ions into the membrane substrate, as the active sites for coordinative reaction, is supposed to be a simple and efficient method. Specifically, the membrane substrate was firstly prepared by blending PES with iron acetylacetonate (Fe(acac)_3_), as the Fe (III) source, via non-solvent induced phase separation (NIPS). The hydrophobic chain of acetylacetonate can intertwist with the PES molecular chain, which increase the stability of Fe(acac)_3_ in the membrane matrix. Subsequently, TA was introduced on the surface and interface of PES/Fe substrate by crosslinking reaction with Fe^3+^, forming the loose selective layer. The thickness of the selective layer was varied with the assembly time. The surface chemical properties, membrane structures and separation performance were evaluated in detail. The optimized PES/Fe-TA membrane displayed satisfactory water permeance, high dye/salt fractionation efficiency and excellent antifouling properties towards dye/salts mixtures. This work provides a facile and scalable production strategy to construct the loose NF membrane, which has great potential for industrial application.

## 2. Materials and Methods

### 2.1. Chemicals and Materials

Polyethersulfone (PES, Ultrason E6020P with M_w_ = 58 kDa) was bought from BASF Co., Ltd. (Shanghai, China).and dried at 110 °C for 12 h before use. Tannic acid (TA, 99%) and iron (III) acetylacetonate (Fe(acac)_3_) were purchased from Aladdin Reagent Co. Ltd. (Shanghai, China). Polyvinylpyrrolidone (PVP, 99%) and N, N-dimethyl formamide (DMF, 99%), Congo Red (CR, 99%), Methyl blue (MB, 99%), Eriochrome Black T (EBT, 99%), Acid Orange74 (AO74, 99%), magnesium chloride (MgCl_2_), magnesium sulfate (MgSO_4_), sodium sulfate (Na_2_SO_4_) and sodium chloride (NaCl) were purchased from Sinopharm Chemical Reagent Co., Ltd. (Shanghai, China).

### 2.2. Membrane Fabrication

The PES/Fe membranes were prepared via non-solvent induced phase separation (NIPS). In detail, PES, PVP and a certain weight of iron acetylacetone (0%, 0.5%, 1.0%, 1.5% and 2.0 wt%) were dispersed in DMF solution and stirred at 60 °C for 6 h to obtain a uniform casting solution. The casting solutions were stored at room temperature for 12 h to ensure a complete release of bubbles and then cast on non-woven fabric using an automated film applicator with a gap of 320 μm. Subsequently, the cast films were immersed into a coagulation bath at room temperature after being exposed to the atmosphere for 20 s. Then, the prepared membranes were immersed in pure water for at least 24 h to leach out the residual solvent before using.

The PES/Fe-TA nanofiltration membranes were prepared via the coordination reaction between TA and Fe. Firstly, the cleaned PES/Fe membranes were immersed in a TA solution (10.0 g/L) and oscillated at different times (0, 1, 2, 3 and 4 h) with a shaker. The PES/Fe-TA nanofiltration membranes were then thoroughly washed with deionized water to remove the unreacted TA. Before testing, the PES/Fe-TA nanofiltration membranes were stored in deionized water.

### 2.3. Membrane Characterization

The chemical structures and elemental compositions of these NF membranes were analyzed by Fourier transform infrared spectroscopy (ATR-FTIR, Nicolet 6700, TMO, Waltham, MA, USA) and X-ray photoelectron spectroscopy (XPS), respectively. The field emission scanning electron microscopy (FESEM, Hitachi S4800, Tokyo, Japan) was operated to characterize the morphology of the NF membranes. The hydrophilicity of these membranes was characterized by a contact angle goniometer (SL-200C, KINO, Boston, MA, USA). The surface zeta potential of membrane was measured by a Sur-PASS electrokinetic analyzer (SurPASS, Anton Paar GmbH, Graz, Austria). Thermogravimetric analysis (TGA, METTLER TGA SF, Mettler Toled, Switzerland) was conducted with a heating rate of 10 °C/min from room temperature to 700 °C under 100 mL/min in air atmosphere.

### 2.4. Filtration Performance

The NF performance of these membranes was tested by a commercial laboratory scale cross-flow flat membrane module with an effective area of 7.065 cm^2^ at room temperature. A schematic diagram of the experimental set up is shown in Figure 1. The membranes were initially compacted for 20 min under 0.3 MPa to obtain a steady permeation and then the pressure was lowered to 0.2 MPa. The water flux (*J*, L·m^−2^·h^−1^) was measured and calculated by the following equation:(1)$$J=\frac{V}{A\times \Delta t}$$
where *V* (L) is the volume of permeated water, A (m^2^) is the effective membrane area and Δ*t* (h) is the permeation time.

The separation performance of these NF membranes was conducted by using 1 g/L salt solution (Na_2_SO_4_, MgSO_4_, MgCl_2_ and NaCl) and 0.1 g/L dye solution (CR, MB, EBT and AO74) as feed solution, respectively. Furthermore, the CR solution (0.1 g/L) mixed with different concentration (2, 4, 6, 8 and 10 g/L) of NaCl solution were also used as feed solution to judge the dye/salt mixture fraction ability. The rejection ratio (*R*) was defined by the following equation:(2)$$R=1-\frac{C_{p}}{C_{f}}$$
where *C_p_* and *C_f_* is the concentration of permeate and feed solution, respectively. Herein, the salt concentration was measured by an electrical conductivity. The dye concentration was measured by a UV-vis spectrophotometer. The maximum absorption wavelength of CR, MB, EBT and AO74 are 488 nm, 664 nm, 410 nm and 484 nm, respectively. All flux and rejection measurements were conducted using three membrane samples.

### 2.5. Antifouling Performance

#### 2.5.1. Static Adsorption Tests

The antifouling measurements of the NF membrane were conducted using CR, MB and EBT as representative pollutants. For the static adsorption tests, the membrane samples (7.56 cm^2^) were immersed in dye solutions (0.1 g/L, *C_i_*) for 2 h. Equilibrium concentrations of dye (*C_e_*) were measured by UV-vis spectrophotometry. The adsorbed mass of dye per unit area of membrane (*Q*, µgcm^−2^) was calculated using Equation (3):(3)$$Q=\frac{\left(C_{i}-C_{e}\right)V}{A}$$
where *A* is the effective membrane area (cm^2^), *V* is the volume of dye solution (mL) and *C_i_* and *C_e_* are the initial and equilibrium dye concentrations (g/L), respectively.

#### 2.5.2. Dynamic Fouling Experiments

In the dynamic antifouling test, 0.1 g/L CR, EBT and MB solutions were used as representative pollutants, respectively. The antifouling filtration experiments were conducted at 0.2 MPa and room temperature. The antifouling test process is as follows: Firstly, the membrane sample was pressurized to reach a stable water flux before the measurement. Then, the pure water flux (*J*_w1_) was continuously measured for 60 min and recorded every 10 min. Afterwards, the membrane filtration was conducted using dye solution as feed solution for another 60 min. The permeate flux of CR, EBT or MB solution (*J_p_*) was also recorded every 10 min. Subsequently, the membrane was cleaned with distilled water for 30 min. Finally, the pure water flux (*J*_w2_) was measured again for 60 min. The water fluxes were calculated by Equation (1).

The antifouling properties was further evaluated by flux recovery ratio (FRR), total fouling ratio (Rt), reversible fouling ratio (Rr) and irreversible fouling ratio (Rir). Those parameters were defined and calculated as follows:(4)$$\mathrm{FRR}=\frac{J_{w2}}{J_{w1}}\times 100\%$$
(5)$$\mathrm{Rt}=\left(1-\frac{J_{p}}{J_{w1}}\right)\times 100\%$$
(6)$$\mathrm{Rr}=\frac{J_{w2}-J_{p}}{J_{w1}}\times 100\%$$
(7)$$\mathrm{Rir}=\left(1-\frac{J_{w2}}{J_{w1}}\right)\times 100\%$$

### 2.6. Long-Term Stability of the Membrane

To evaluate the long-term stability of optimized NF membranes, the CR (0.1 g/L) solutions mixed with NaCl (2 g/L) were used as feed solution to filtrated for 24 h. The permeate flux and rejections for CR and NaCl were monitored by the aforementioned methods.

## 3. Results and Discussion

### 3.1. Chemical Structure and Properties of Membranes

The fabrication process of LNM combined blending and interfacial coordination strategy was depicted in Figure 2. For the control, the pristine PES membrane was also fabricated using the NIPS technique. The photographs of the PES, PES/Fe and PES/Fe-TA membrane are shown in Figure 3a. It can be observed that the pristine PES membrane shows a white color and the PES/Fe membrane exhibits an orange color. The color change is ascribed to the color of iron acetylacetonate, suggesting the successful incorporation of Fe^3+^ on the membrane matrix. After the immersion of TA solution, the PES/Fe-TA membrane displays dark grey, demonstrating the TA coating is assembled on the membrane surface. In order to further confirm the existence of Fe and TA on PES membrane, the TGA and FTIR analysis were studied. Figure 3b exhibits the results of TGA analysis for the PES, PES/Fe and PES/Fe-TA membrane. The residual weights of PES/Fe and PES/Fe-TA membrane were 9.8% and 10.2%, while the pristine PES membrane was completely burned out in the air atmosphere. The increase of residual mass should correspond to the iron base compound in the PES/Fe and PES/Fe-TA membranes, providing further evidence of the iron complex loading. FT-IR was employed to analyze the surface chemical structure of PES, PES/Fe and PES/Fe-TA membranes, as shown in Figure 3c. The absorption peaks at 1150 cm^−1^ and 1296 cm^−1^ are the symmetric and asymmetric stretching vibrational peaks of the S=O functional group in PES. The stretching vibration peak between benzene ring and S in PES is located at 1100 cm^−1^. Compared with the PES and PES/Fe membrane, an additional adsorption band at 1720 cm^−1^ can be observed in the spectrum of the PES/Fe-TA membrane. This can be ascribed to the C=O stretching vibrations of the of TA [39], suggesting that the TA were successfully incorporated on PES/Fe membrane surface.

Hydrophilicity of membrane is a vital parameter affecting membrane permeability and antifouling performance during filtration applications [40]. The surface hydrophilicity of the pristine PES, PES/Fe and PES/Fe-TA membranes was evaluated by dynamic water contact angle (WCA) measurements, and the results are shown in Figure 4a. The pristine PES membrane exhibited high hydrophobicity with the initial WCA of 75°, and almost remained unchanged within 60 s. For the PES/Fe membrane, the WCA is slightly higher than that of the PES membrane, which is ascribed to the low surface energy of Fe(acac)_3_. However, the initial WCA of PEA/Fe-TA membrane decreased to around 49.8° and dramatically declined to 27.5° in 60 s, indicating the improved hydrophilicity after TA assembly. This can be attributed to the hydrophilic phenolic hydroxyl groups formed on the surface of the PES/Fe-TA membrane. Moreover, the surface charge also plays a significant role in the separation properties of membranes. The surface charge of PES, PES/Fe and PES/Fe-TA membrane are studied by the surface zeta potential (Figure 4b). It can be seen that PES/Fe-TA membrane displays enhanced negative charge compared with PES and PES/Fe membranes. That is because TA had many phenolic hydroxyl groups which could release hydrogen ions to endow membrane surfaces with negative charge. As the pH increases, more phenolic hydroxyl groups deprotonate, resulting in a stronger negative charge.

### 3.2. Effects of TA Assembly Time on the Membrane Structure and Performance

The reaction time is an important factor for TA deposition on the PES/Fe substrate. To regulate the TA layer in a controllable thickness, the effects of assembly time on the membrane structure and performance were studied. Due to the instability of Fe(acac)_3_ in ethanol, the PES/Fe sample has not been dried by supercritical drying apparatus, which means that its morphologies cannot be shown for control. The morphologies of PES/Fe-TA membranes treated at different TA assembly times were inspected by SEM, as shown in Figure 5. The membranes with TA coordination assembly have a flat and smooth surface. Since the interaction between Fe^3+^ in membrane matrix and TA can effectively suppress the TA-Fe complex particles on the membrane surface, a relatively smooth surface was observed on the PES/Fe-TA membranes. In addition, many pores (pore size of 10–50 nm) were observed on the surface of PES/Fe-TA1h membrane and these pores became lesser and smaller with the increase of TA assembly time (Figure 5a–d). This result was attributed to the assembly of TA on the PES/Fe substrate, forming a uniform Fe-TA complex layer on the membrane surface and reducing the pore sizes.

The cross-section images in Figure 5e–h show that a top layer formed on the PES/Fe support after TA assembly. Moreover, the thickness of the top layer increased from ~30 nm to 119 nm with the increase of the TA assembly time. The Fe^3+^ migrated to the membrane surface and coordinated with TA to form the thicker separation layer with the increase of the TA assembly time. Therefore, the interfacial coordination of polyphenolic layer can form a smooth surface and controllable thickness of separation layer by varying the TA assembly time, which might decide the separation performance of the membrane.

The pure water flux and CR rejection of PES/Fe-TA membranes with different TA assembly times is shown in Figure 6. It can be seen that the pure water permeance of the PES/Fe-TA membrane gradually decreased, while the rejections of CR increased with the increase of assembly time. The water permeance decreased from 305.2 L·m^−2^·h^−1^ to 133.6 L·m^−2^·h^−1^, and the CR rejection increased from 89.7% to 98.1%. These results could be attributed to the fact that a thicker separation layer was formed on the membrane surface with the increase of TA assembly time, leading to the smaller pore size and increased permeation resistance, as shown in Figure 5. Comprehensively considering the water flux and rejection, the assembly time was fixed to 3 h for the following tests. For the PES/Fe-TA3h membrane, the water permeance reached 175.8 L·m^−2^·h^−1^ and the rejection rate of CR was 97.7% at 0.2 MPa.

### 3.3. Nanofiltration Performance

The nanofiltration properties of PES/Fe-TA3h membrane were measured by using four dyes (CR, MB, EBT and AO74) and four inorganic salts (Na_2_SO_4_, MgSO_4_, MgCl_2_ and NaCl). The flux and rejections of the PES/Fe-TA3h membrane for different dyes and inorganic salts were measured in a cross-flow filtration apparatus under 0.2 MPa. Figure 7a,b presented the results of nanofiltration performance for a single component of dye (0.1 g/L) or salt (2 g/L). The rejections to CR, MB, EBT and AO 74 was 97.7%, 97.1%, 95.0% and 58.6%, while the fluxes were 80.1, 70.0, 93.5 and 133.5 L·m^−2^·h^−1^, respectively. The difference of rejections for the dyes may be ascribed to the molecular size. The relative molecular weight of AO 74 was 417.35 g/mol, which is lower than 696.66, 799.80 and 461.38 g/mol for CR, MB and EBT; thus, they can more easily pass through the membrane pores when permeating the membrane. The retentions of Na_2_SO_4_, MgSO_4_, NaCl and MgCl_2_ was 6.2%, 4.9%, 2.6% and 1.9% respectively, which conforms to the typical negatively charged NF membranes (Figure 7b). The rejections of dyes and inorganic salts are decided with the coaction of steric and Donnan effects [41,42]. For this result, the PES/Fe-TA membrane is only used for dyes with a molecular weight of a least 700 Da. The SO_4_^2−^ has stronger repulsive interaction with membrane surface than that with Cl^−^, resulting in higher rejections for Na_2_SO_4_ and MgSO_4_. The high rejections of dyes may be ascribed to the presence of a hydration shell of charged dye molecules and/or aggregates of dye molecules with a size of tens of nanometers. However, the hydrate radius (<0.5 nm) of salts is much smaller than the membrane pore size, due to the dominant role of sieve principle, and the rejection of salts was low. High dye rejection ability and weak salt rejection ability proved that the prepared PES/Fe-TA membrane can be applied to dye desalination.

It is believed that the presence of salt in the dye solution has some effects on the membrane separation performance; therefore, the separation properties of dye and salt mixture was investigated. The 0.1 g/L CR with different NaCl content were used to form different salinity feed solution. The results of permeate flux and solute rejections for the CR/NaCl mixture are shown in Figure 7c. It can be seen that the permeability of CR/NaCl mixture solution decreased from 79.6 L·m^−2^·h^−1^ to 64.8 L·m^−2^·h^−1^. Moreover, the rejections of CR and NaCl also decreased, from 97.6% to 96.0% and from 2.5% to 1.45% respectively, with the increase of NaCl concentration (from 2 g/L to 10 g/L). This was because salt tends to disperse dye molecules uniformly in the mixed solution and avoid the aggregation of dye molecules, resulting in smaller dye particles to permeate through the membrane pores. Meanwhile, the dye adsorbed on the membrane pores and the concentration polarization occurred on the membrane surface, which resulted in decreased permeability and dye rejection.

In order to highlight the prominent properties of the membranes prepared in this work, we compared the water flux, dye and salt rejections of PES/Fe-TA3h membrane with those reported in other research (Table 1). It can be seen that the LNM prepared in this study showed good dye/salt separation capability compared to the results reported in the selected literature.

### 3.4. Antifouling Properties

Membrane fouling is one of trickiest problems in membrane processes and it results in many drawbacks, such as permeance decline, increase in operational costs and membrane degeneration. The TA-Fe complex was super-hydrophilicity and expected to overcome the fouling problem of nanofiltration membrane for the separation dye solution. The antifouling property of the PES/Fe-TA membrane was evaluated with static adsorption and dynamic filtration experiments using CR, EBT and MB as the model dye pollutants. The results of static adsorption experiments for three dyes on PES, PES/Fe and PES/Fe-TA membranes are shown in Figure 8. It can be seen that the PES/Fe-TA membrane has the smallest adsorption capacity for the three dyes, compared with PES and PES/Fe membranes. The adsorption amounts of CR, EBT and MB are, respectively, 0.018, 0.026 and 0.043 mg/cm^2^ for the PES/Fe-TA membrane compared to 0.25, 0.22 and 0.33 mg/cm^2^ for the pristine PES membrane. Figure 8b shows the surface colors of PES, PES/Fe and PES/Fe-TA3h membrane with static adsorption of the three dyes. We can clearly see that the membrane colors have changed to the dye color after adsorption. Moreover, the adsorption behavior for the PES and PES/Fe membrane are more serious than for the PES/Fe-TA membrane. This phenomenon was attributed to the corporation of the hydrophilicity and the charge repulsion to negative dyes. The TA on the PES/Fe-TA membrane make it more difficult for the dyes to adhere to the membrane surface, which can enhance the dye pollution resistance in the filtration dye solution.

In order to further evaluate the antifouling property of PES/Fe-TA membrane, the dynamic cyclic filtration experiment with different dye solutions was conducted and the results are revealed in Figure 9a. It can be seen that the permeate flux of the dye solution are lower than the pure water. This is probably caused by the dye fouling and concentration polarization. Moreover, the water flux recovers by the water/ethanol dilute solution cleaning treatment after each cycle of dye filtration test. In addition, the values of *J_w1_*, *J_p_* and *J_w2_* measured in the two cycles were used to calculate FRR, Rt, Rr and Rir to evaluate the anti-fouling properties of the PES/Fe-TA membrane, as shown in Figure 9b. After two cycles, the FRR values of CR, EBT and MB were 90.3%, 87.5% and 81.6%, respectively, and the corresponding R_t_ values were 62.6%, 57.3% and 65.7%, respectively. It shows that the PES/Fe-TA3h membrane has excellent antifouling property on CR, EBT and MB. Based on the above results, the excellent antifouling ability of PES/Fe-TA3h membrane promotes its further application in dye desalination and dye wastewater treatment. These results clearly indicated that TA, indeed, acted as a strong dye adsorption resistance.

### 3.5. Long-Term Stability of the PES/Fe-TA Membrane

In order to explore the long-term stability of the membrane, the dye/salt fractionation performance of the PES/TA-Fe3h membrane was investigated with 24 h continuous filtration of the mixed solution (0.1 g/L CR and 2 g/L NaCl). As shown in Figure 10, the permeate flux slightly declined from 77.0 to 57.0 L·m^−2^·h^−1^ in the first few hours, which could be ascribed to the adsorption of CR and evolution of a dye cake layer on the membrane surface during the filtration. Moreover, the permeate flux stabilizes at 49.0 L·m^−2^·h^−1^ when the adsorption reaches dynamic equilibrium. Meanwhile, the rejections of CR and NaCl were increased slightly at the beginning, and then reached stability (>97.2% to CR and <5.4% to NaCl). The results confirmed that PES/Fe-TA3h membrane exhibited a long-term stability, which has great potential to be used in dye desalination and dye wastewater treatment.

## 4. Conclusions

In this study, a polyphenol-based loose nanofiltration membrane was successfully developed by integrating blending and interfacial coordination strategy. The iron ion complex was introduced in PES membrane via NIPS method and TA coordinated with Fe^3+^ forming the loose separation layer. The thickness of TA layer was controlled by altering the TA assembly time. The introduction of TA improved the surface hydrophilicity and negative charge. The optimized membrane with assembly 3 h (PES/Fe-TA3h) showed a pure water flux of 175.8 L·m^−2^·h^−1^, dye rejections of 97.7%, 97.1% and 95.0% for CR, MB and EBT, respectively, along with salt penetration rates of 93.8%, 95.1%, 97.4% and 98.1% for Na_2_SO_4_, MgSO_4_, NaCl and MgCl_2_, respectively, at 0.2 MPa. Moreover, the PES/Fe-TA3h membrane exhibited a stable dye rejection and salt permeation in the 24 h continuous test and high FRR of 90.3%, 87.5% and 81.6% for CR, EBT and MB, respectively. This study provides a new way to fabricate the loose nanofiltration membrane for effective fractionation of dyes and salts in the saline textile wastewater.

## Figures and Tables

**Figure 1 membranes-12-00340-f001:**
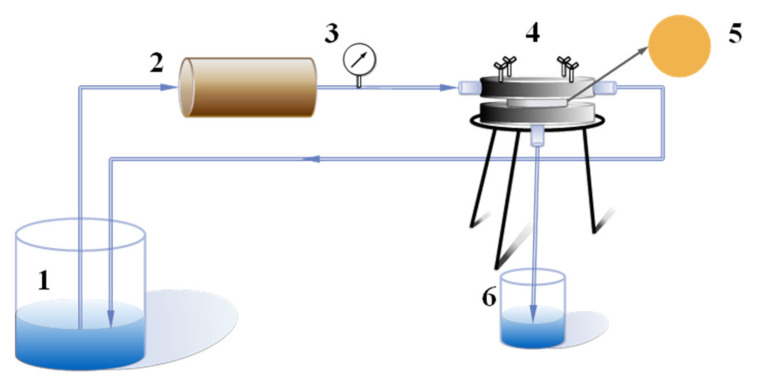
Schematic diagram of cross-flow experimental device: 1. Feed liquid; 2. Peristaltic pump; 3. Pressure gauge; 4. Membrane assembly; 5. Measured film; 6. Penetrating fluid.

**Figure 2 membranes-12-00340-f002:**
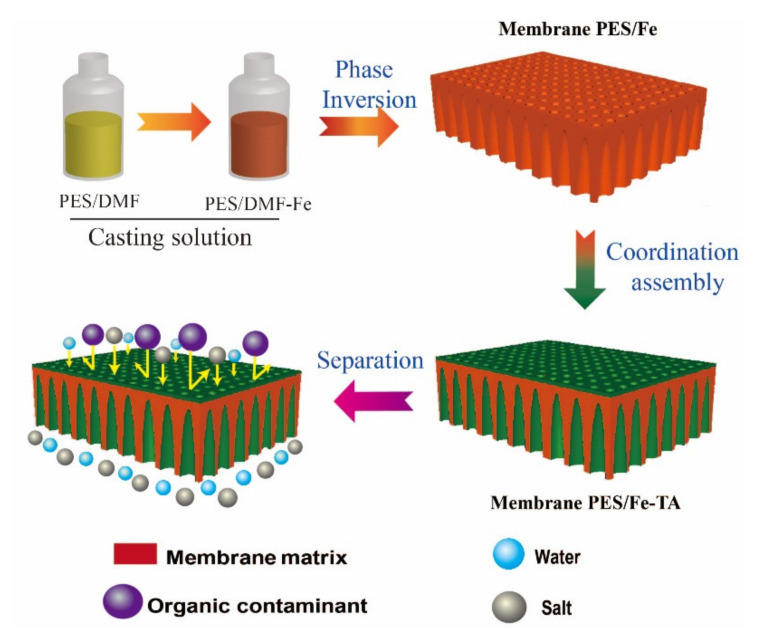
Schematic representation of the fabrication process of loss nanofiltration membrane and selective separation of dye and salt.

**Figure 3 membranes-12-00340-f003:**
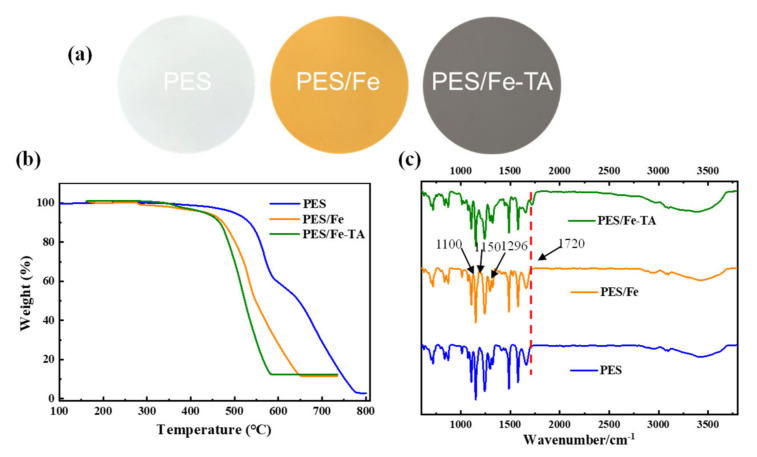
(**a**) Digital photographs of membrane surface, (**b**) TGA curves and (**c**) ATR-FTIR spectra of the PES, PES/Fe and PES/Fe-TA membrane.

**Figure 4 membranes-12-00340-f004:**
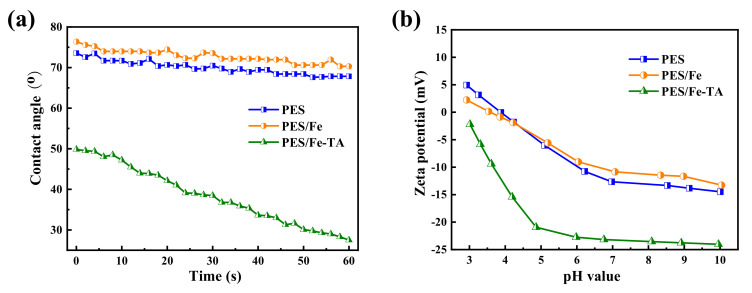
(**a**) The water contact angle and (**b**) zeta potentials of the PES, PES/Fe and PES/Fe-TA membranes.

**Figure 5 membranes-12-00340-f005:**
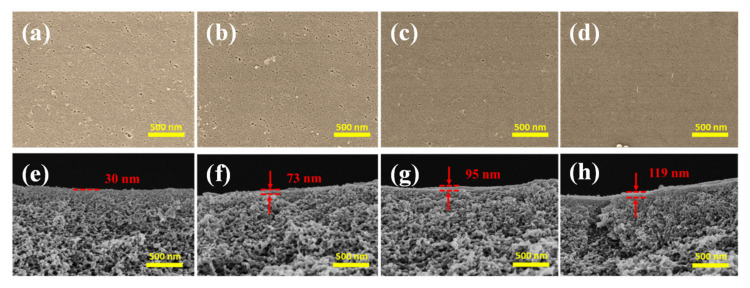
The surface (**a**–**d**) and cross section morphology (**e**–**h**) of PES/Fe-TA membranes with different TA assembly times.

**Figure 6 membranes-12-00340-f006:**
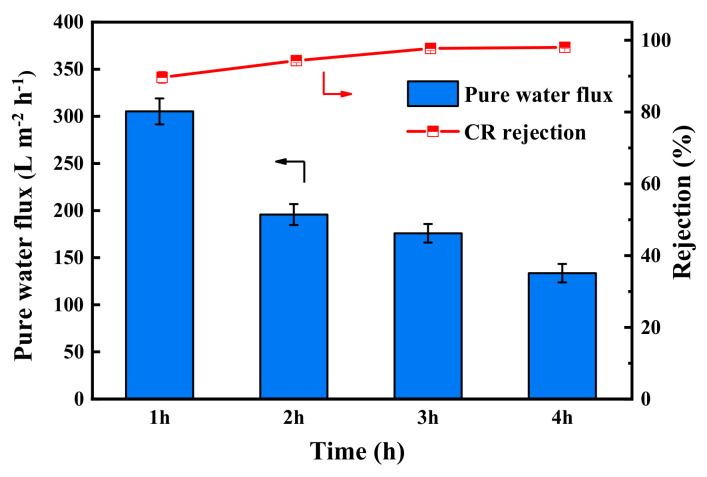
The pure water flux and CR rejection of PES/Fe-TA membrane at different TA assembly times.

**Figure 7 membranes-12-00340-f007:**
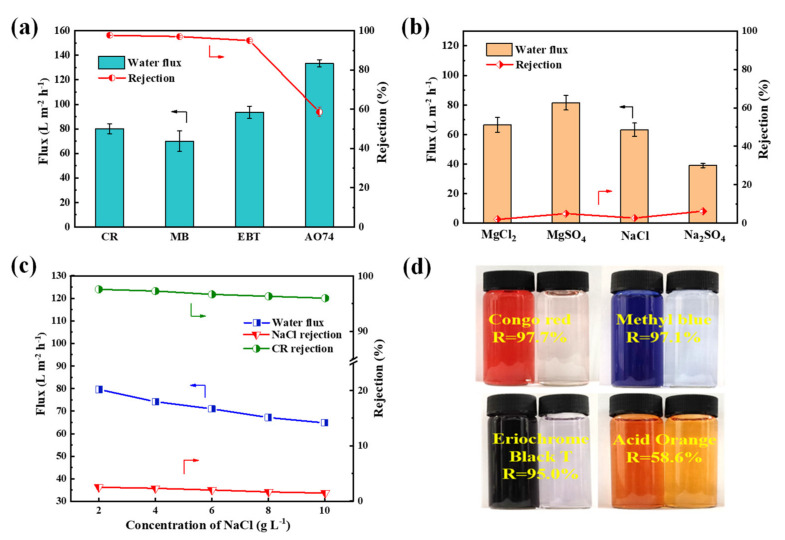
(**a**) Filtration performance of single dye solution, (**b**) rejections of salts, (**c**) filtration performance for the dye/salt mixture solution for the PES/Fe-TA3h membrane and (**d**) photographs of feed and penetration solutions.

**Figure 8 membranes-12-00340-f008:**
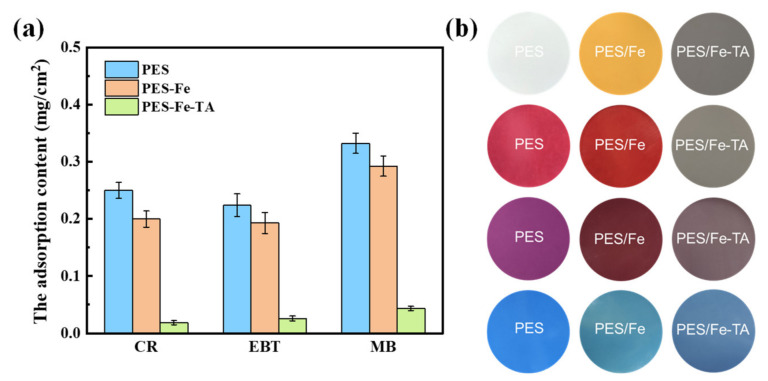
The dye adsorption content (**a**) and digital photos of surface color (**b**) on PES, PES/Fe and PES/Fe-TA3h membranes with static adsorption for different dyes.

**Figure 9 membranes-12-00340-f009:**
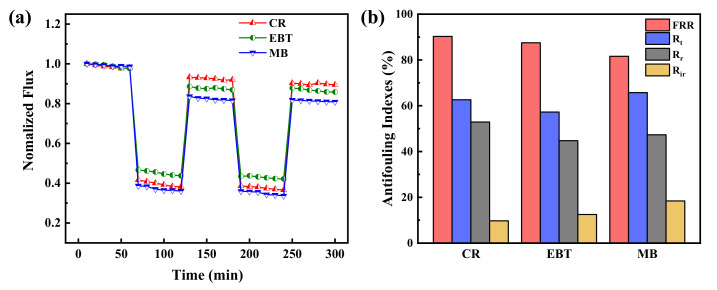
(**a**) The time-dependent normalized flux during the filtration of CR, EBT and MB solution and (**b**) antifouling indexes for the PES/Fe-TA3h membrane.

**Figure 10 membranes-12-00340-f010:**
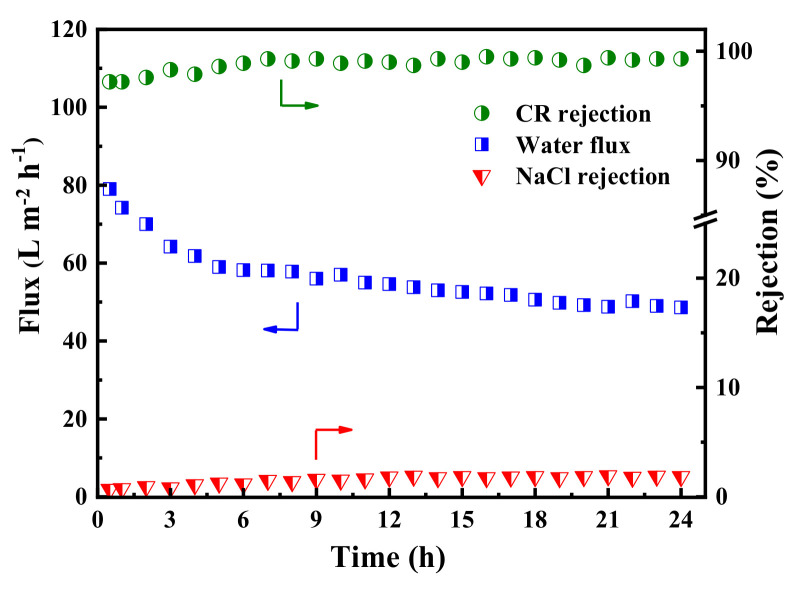
The long-term operation stability of the PES/Fe-TA3h membrane for the CR/NaCl mixture solution (feed: 0.1 g/L CR and 2 g/L NaCl).

**Table 1 membranes-12-00340-t001:** Comparison of the performance of the NF membranes in the literature.

Membranes	Permeate Flux (L m^−2^ h^−1^)	Congo Red		NaCl		Pressure (MPa)	Ref.
		C(g/L)	R(%)	C(g/L)	R(%)		
**TA/GOQDs-1**	**23.3**	0.1	99.8	1	17.2	0.2	[43]
TiO_2_-HMDI	30.5	0.035	97.4	1	2.7	0.2	[44]
PSF/GO	73.7–95.4	0.1	99.9	1	<5	0.2	[45]
PAN-PEI-GA	51.0	0.1	97.1	1	5	0.2	[46]
PAN-DR80	113.6	0.1	99.8	1	12.4	0.4	[47]
Fe(III)-phos-(PEI)/HPAN	12.1	0.1	99.5	1	7.5	0.2	[48]
CaCO_3_/PEI-GA	141	0.1	99.6	1	6.9	0.3	[49]
PST-1	52.3	0.1	99.0	1	<7	0.6	[50]
TAIP M4	31.5	0.2	99.4	2	5.4	0.1	[51]
PDA/SBMA/HPAN	68.8	0.5	98.2	1	5.0	0.4	[52]
LNFM-2	212.9	0.2	99.6	1	5.6	0.4	[25]
PES/Fe-TA3h	77.0	0.1	97.7	2	2.6	0.2	This work

## Data Availability

Not applicable.

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
