# Peer review of "Metal-Coordinated Nanofiltration Membranes Constructed on Metal Ions Blended Support toward Enhanced Dye/Salt Separation and Antifouling Performances"

_membranes, 2022, doi:10.3390/membranes12030340_

Round 1
Reviewer 1 Report
The manuscript is technically good. I recommend to the authors to cite recent reviews, books and book chapters.
Author Response
Reviewer #1
Comment: The manuscript is technically good. I recommend to the authors to cite recent reviews, books and book chapters.
Reply: We thank Reviewer1 very much for the positive comment. Following reviewer’s suggestion, the corresponding literatures have been added in revised manuscript as follows:
Page 14
“[11] Peng L E, Yang Z, Long L, et al. A critical review on porous substrates of TFC polyamide membranes: mechanisms, membrane performances, and future perspectives[J]. J. Membr. Sci., 2022, 641: 119871.
[12] Sajid M, Sajid Jillani S M, Baig N, et al. Layered double hydroxide-modified membranes for water treatment: recent advances and prospects[J]. Chemosphere, 2022, 287: 132140.
[13] Tijing, L.D.; Woo, Y.C.; Yao, M.; Ren, J. Electrospinning for Membrane Fabrication: Strategies and Applications. In Comprehensive Membrane Science and Engineering; Elsevier: Oxford, UK, 2017; ISBN 9780124095472.”

Reviewer 2 Report
This study provides a novel and elegant method to fabricate a loose nanofiltration membrane via integrating blending and interfacial coordination strategy. The obtained membranes show enhanced antifouling resistance and simultaneous dye/salt separation performance. The research object of this work is of great significance in the membrane separation field. The idea is novel and interesting, and it also provides valuable information for water treatment. Therefore, it can be accepted for publication after minor revision.
- To make the study clearly, the detailed description and reason for iron acetylacetonate should be provided.
- It is suggested to keep the style of the symbols in the formula (4) uniform. Besides, the size of the font is recommended to be uniform in Fig. 7 (b).
- In section 3.3, the nanofiltration properties of PES/Fe-TA3h membrane were measured with four dyes (CR, MB, EBT, and AO 74). Compared to the other three, the rejection of AO 74 was low. It should be given the corresponding discussion and explanation.
Author Response
Reviewer #2
Comment: This study provides a novel and elegant method to fabricate a loose nanofiltration membrane via integrating blending and interfacial coordination strategy. The obtained membranes show enhanced antifouling resistance and simultaneous dye/salt separation performance. The research object of this work is of great significance in the membrane separation field. The idea is novel and interesting, and it also provides valuable information for water treatment. Therefore, it can be accepted for publication after minor revision.
Reply: We thank reviewer 2 very much for the comments and suggestions. We have carried out the following point-by-point revisions based on the reviewer’s comments and suggestions.
- To make the study clearly, the detailed description and reason for iron acetylacetonate should be provided.
Reply: Thanks for the helpful comment from Reviewer 2. The corresponding revision has been added in the introduction part as follows:
Page 2“The hydrophobic chain of acetylacetonate can intertwist with the PES molecular chain, which increase the stability of Fe(acac)3 in membrane matrix.”
2.It is suggested to keep the style of the symbols in the formula (4) uniform. Besides, the size of the font is recommended to be uniform in Fig. 7 (b).
Reply: The corresponding revision have been made in the revised manuscript as follows:
Page 5
(4)
(5)
(6)
(7)
Page 11
Figure 8. The dye adsorption content (a) and surface color (b) on PES, PES/Fe and PES/Fe-TA3h membranes with static adsorption for different dyes.
(Since we have added a new device diagram to the article, the serial number of Figure 7 has become Figure 8)
3.In section 3.3, the nanofiltration properties of PES/Fe-TA3h membrane were measured with four dyes (CR, MB, EBT, and AO 74). Compared to the other three, the rejection of AO 74 was low. It should be given the corresponding discussion and explanation.
Reply: We appreciate the helpful comments from Reviewer 2. Following Reviewer’s suggestion, the discussion has been revised as follows:
Page 8 “ The relative molecular weight of AO 74 was 417.35 g/mol, which is lower than 696.66, 799.80 and 461.38 g/mol for CR, MB and EBT, resulting more easily to pass through the membrane pores when permeating the membrane.”

Reviewer 3 Report
General comments and summary
In this study, authors synthesize membrane with the application in reverse osmosis technology.
The experimental work is scientifically sound; however, the manuscript can be further improved. Please refer to my comments below in the specific comment section
Specific comments
- PES, CR, MB and EBT on first mentioning should not be the abbreviation.
- Section 2.1 Do mention the grade of the reagents, and the purity of the dyes.
- Do mention the wavelength for measuring the dyes using uv/vis
- Are there any coagulation of dyes after adding the salt solution? CR is known to coagulate in saline solution
- Section 2.4 Authors should include schematic diagram of the experimental set up,
- Symbol used in Equation 2 should be consistent with the symbols described in line 151. Do subscript the p and the f in eq 2. Do the same with the symbols in line 162-163. Do recheck for whole manuscript to ensure consistency of symbols.
- Fig 2a. What are digital photographs? Are these the surface colours? Fig 2a shows three circles of different colours. Authors need to be informative with the figure caption
- Fig 2c. FTIR peaks should be marked clearly. The description and interpretation of the FTIR data seem minimal and need to be improved.
- Is hydrophilic membrane more prefer than hydrophobic membrane? Will it hydrophilicity speed up fouling of membrane?
- Fig 7a. Why are the same PES change colour? Are these colours real surface colour or colour from computer?
Author Response
General comments and summary
In this study, authors synthesize membrane with the application in reverse osmosis technology. The experimental work is scientifically sound; however, the manuscript can be further improved. Please refer to my comments below in the specific comment section.
Reply: We thank reviewer 3 very much for the comments and suggestions. We have carried out the following point-by-point revisions based on the reviewer’s comments and suggestions.
Specific comments
1.PES, CR, MB and EBT on first mentioning should not be the abbreviation.
Reply: We thank Reviewer 3 for the helpful comment. The corresponding revision have been made in the revised manuscript as follow:
Page1“Specially, iron acetylacetonate was firstly blended in Polyethersulfone (PES) substrate via non-solvent induced phase separation (NIPS)…dye rejections of 97.7 %, 97.1% and 95.0% for Congo red (CR), Methyl blue (MB) and Eriochrome Black T (EBT),…”
2.Section 2.1 Do mention the grade of the reagents, and the purity of the dyes.
Reply: The grade of the reagents, and the purity of the dyes have been elucidated in the revised manuscript as follow:
Page 3“Tannic acid (TA,99%) and iron (III) acetylacetonate (Fe(acac)3) were purchased from Aladdin Reagent Co. Ltd. Polyvinylpyrrolidone (PVP,99%) and N, N-dimethyl formamide (DMF,99%), Congo Red (CR,99%), Methyl blue (MB,99%), Eriochrome Black T (EBT,99%), Acid Orange74 (AO74,99%)”.
3.Do mention the wavelength for measuring the dyes using uv/vis
Reply: We thank the suggestion from Reviewer. The wavelength for measuring the dyes have been added in the revised manuscript as follow:
Page 4“The maximum absorption wavelength of CR, MB, EBT and AO74 are 488 nm, 664 nm, 410 nm and 484 nm, respectively.”
4.Are there any coagulation of dyes after adding the salt solution? CR is known to coagulate in saline solution.
Reply: We thank the question from Reviewer. In the course of our experiments no coagulation of dyes. It was believed that dye molecules were likely to develop clusters in aqueous solutions and adding salts can form a strong charge layer around the dye, reducing the coalescence of dye particles in the water would lessen this agglomeration so that dyes were dispersed uniformly [Sep. Purif. Technol. 129 (2014)96–105.]. About this situation, we will research it in the future study.
5.Section 2.4 Authors should include schematic diagram of the experimental set up.
Reply: The suggestion from Reviewer is helpful. The schematic diagram of the experimental set up have been added in the revised manuscript as follow:
Page 4
Figure.1 Schematic diagram of cross-flow experimental device: 1. feed liquid; 2. Peristaltic pump; 3. pressure gauge; 4. membrane assembly; 5. measured film; 6. penetrating fluid
6.Symbol used in Equation 2 should be consistent with the symbols described in line 151. Do subscript the p and the f in eq 2. Do the same with the symbols in line 162-163. Do recheck for whole manuscript to ensure consistency of symbols.
Reply: We appreciate the comments from Reviewer. The description of symbols has been revised in the manuscript as follow:
Page 4 “
(2)
where Cp and Cf was the concentration of permeate and feed solution, respectively.”
And the manuscript has been carefully checked and revised.
7.Fig 2a. What are digital photographs? Are these the surface colours? Fig 2a shows three circles of different colours. Authors need to be informative with the figure caption.
Reply: The figures of Fig2a are photographs of the real membrane surface. The corresponding revision has been added in the manuscript as follow:
Page 6“Figure 3. (a) Digital photographs of membrane surface, (b) TGA curves and (c) ATR-FTIR spectra of the PES, PES/Fe, and PES/Fe-TA membrane.”
(Since we have added a new device diagram to the article, the serial number of Figure 2 has become Figure 3)
8.Fig 2c. FTIR peaks should be marked clearly. The description and interpretation of the FTIR data seem minimal and need to be improved.
Reply: We thank Reviewer for the comments. The corresponding description has been added in revised manuscript as follows:
Page 6
Figure 3. (a) Digital photographs of membrane surface, (b) TGA curves and (c) ATR-FTIR spectra of the PES, PES/Fe, and PES/Fe-TA membrane.
“The absorption peaks at 1150 cm-1 and 1296 cm-1 are the symmetric and asymmetric stretching vibrational peaks of the S=O functional group in PES. The stretching vibration peak between benzene ring and S in PES is located at 1100 cm-1. Compared with the PES and PES/Fe membrane, an additional adsorption band at 1720 cm-1 can be observed in the spectrum of the PES/Fe-TA membrane. This can be ascribed to the C=O stretching vibrations of the of TA[39], suggesting that the TA were successfully incorporated on PES/Fe membrane surface.”
9.Is hydrophilic membrane more prefer than hydrophobic membrane? Will it hydrophilicity speed up fouling of membrane?
Reply: We thank Reviewer for the comments. In general, the hydrophobic membranes are very susceptible to pollutant fouling with adsorption. By contrast, hydrophilic separation membranes usually have good hydration, and the hydration layer formed in water can effectively prevent contaminants from direct contact and adhesion with the separation membrane surface, thus reducing membrane contamination [Chem. Soc. Rev., 2022: 10.1039].So It is generally believed that hydrophilic membrane corresponds to lower membrane fouling potential than hydrophobic one.
10.Fig 7a. Why are the same PES change colour? Are these colours real surface colour or colour from computer?
Reply: The digital photos are the color of the membrane surface after filtering the dyes, which reflects the degree of adsorption of different dyes on three membranes, also indicates the enhanced dye pollution resistance in the filtration dye solution for the membrane with TA modification.

Reviewer 4 Report
This manuscript includes the strategy to fabricate a loose nanofiltration membrane via interfacial coordination strategy. Iron acetylacetonate was blended in PES substrate and then the loose selective layer was formed by crosslinking reaction between tannic acid (TA) and iron. There are some questions should be revised to reach the publication quality and to increase the satisfaction of readers for the MDPI membranes journal.
- In Figure 1, how does iron acetylacetonate exist in PES after blending? Is there any network between iron and PES?
- Please clearly explain the reaction mechanism between TA and Fe on the surface PES/Fe. Much Fe molecules are necessary in order to form the thin TA film. Is it possible to provide much Fe from the surface of PES/Fe?
- There is no evidence of the coordination of TA and Fe.
- In Figure 2, TGA results showed that the residual weights of PES/Fe and PES/Fe-TA membrane 198 were 9.8% and 10.2 %. Does it mean that PES membrane contains about 10% of Fe?
Author Response
Comment: This manuscript includes the strategy to fabricate a loose nanofiltration membrane via interfacial coordination strategy. Iron acetylacetonate was blended in PES substrate and then the loose selective layer was formed by crosslinking reaction between tannic acid (TA) and iron. There are some questions should be revised to reach the publication quality and to increase the satisfaction of readers for the MDPI membranes journal.
Reply: We appreciate reviewer 4 very much for the comments. We have carried out the following point-by-point revisions based on the reviewer’s comments.
- In Figure 1, how does iron acetylacetonate exist in PES after blending? Is there any network between iron and PES?
Reply: We thank Reviewer for the comments. The presence of iron acetylacetonate can be demonstrated by different film color changes and TGA results.
Since iron acetylacetonate (Fe(C5H7O2)3) has -C5H7O2- as hydrophobic chain, it is able to entangle and interact with hydrophobic PES molecular chains, increasing the stablility of Fe in the membrane matrix .
2.Please clearly explain the reaction mechanism between TA and Fe on the surface PES/Fe. Much Fe molecules are necessary in order to form the thin TA film. Is it possible to provide much Fe from the surface of PES/Fe?
Reply: The iron acetylacetonate, as the Fe (III) source, was introduced in membrane via non-solvent induced phase separation method. And the Fe were distributed on the membrane substrate. When the PES/Fe ultrafiltration membrane is immersed in acidic TA solution, the Fe on the membrane surface will coordinate with TA to form a stable mesh structure, and the Fe in the matrix may also migrate to the membrane interface to coordinate with TA and build a Fe-TA dense separation layer. The content and distribution of Fe in membrane matrix are the key points. The selection of the iron acetylacetonate in this work is the important innovation for fabricating the PES/Fe-TA loose NF membrane.
3.There is no evidence of the coordination of TA and Fe.
Reply: We thank Reviewer for the comments. Three galloyl groups from TA can react with each Fe (III) ion to form a stable octahedral complex which has been reported in other literature [J. Colloid Inter. Sci. 505 (2017) 642–652]. For the direct characterization of Fe-TA complex will further be studied in future work.
4.In Figure 2, TGA results showed that the residual weights of PES/Fe and PES/Fe-TA membrane 198 were 9.8% and 10.2 %. Does it mean that PES membrane contains about 10% of Fe?
Reply: Reviewer 4’s comment is correct. The residual weights of PES/Fe and PES/Fe-TA membrane were 9.8% and 10.2 %, while the pristine PES membrane was completely burnt out in the air atmosphere. The 9.8% and 10.2 % of residual mass should correspond to the iron base compound, such as Fe3O4, which provided further evidence of the iron complex loading in the membrane matrix.

Round 2
Reviewer 4 Report
There is no further comments.
Thanks.